# Towards generalizing deep-audio fake detection networks

**Konstantin Gasenzer, Moritz Wolter**  *{konstantin.gasenzer, moritz.wolter}@uni-bonn.de*
*High Performance Computing and Analytics Lab, University of Bonn, Germany*

**Reviewed on OpenReview:** *https://openreview.net/forum?id=RGewtLtvHz*

## Abstract

Today's generative neural networks allow the creation of high-quality synthetic speech at scale. While we welcome the creative use of this new technology, we must also recognize the risks. As synthetic speech is abused for monetary and identity theft, we require a broad set of deepfake identification tools. Furthermore, previous work reported a limited ability of deep classifiers to generalize to unseen audio generators. We study the frequency domain fingerprints of current audio generators. Building on top of the discovered frequency footprints, we train excellent lightweight detectors that generalize. We report improved results on the WaveFake dataset and an extended version. To account for the rapid progress in the field, we extend the WaveFake dataset by additionally considering samples drawn from the novel Avocodo and BigVGAN networks. For illustration purposes, the supplementary material contains audio samples of generator artifacts.

## 1 Introduction

The advancement of generative machine learning enables digital creativity, for example, in the form of more immersive video games and movies. However, it also creates new digital ways to lie. The technology is abused for theft (Karimi, 2023; Khatsenkova, 2023) and disinformation (Satariano & Mozur, 2023). Scammers use audio fakes to apply for remote jobs illicitly (PC-Mag, 2023). Via the telephone, cloned voices are misused in attempts to trick unsuspecting family members and stage fake kidnappings (Karimi, 2023). Problems have also surfaced on large platforms. Recently, artificially generated songs with voices from two well-known artists illicitly appeared on a large music streaming service (Guardian, 2023). The fake recordings had been created and published without the artist's consent. Consequently, we must meet the advancements in generative machine learning with deep fake detection tools that generalize well.

Our study reveals stable frequency domain artifacts for many modern speech synthesis networks. We visualize generator artifacts for all generators in the WaveFake dataset (Frank & Schönherr, 2021) generators and the Avocodo (Bak et al., 2022) and BigVGAN (Lee et al., 2023a) networks.

We challenge the commonly held belief that deep networks do not generalize well to unknown generators in the audio domain (Frank & Schönherr, 2021). Our results show that deep networks indeed generalize well. Using our dilated convolution-based model, we observe generalization to unseen generators for networks trained on Wavelet Packet Transform (WPT) and Short-Time Fourier Transform (STFT) inputs. We reproduce and improve upon synthetic media-recognition results published for the WaveFake dataset (Frank & Schönherr, 2021).

To ensure our detectors identify the newest generators, we extend the dataset proposed by Frank & Schönherr (2021) by adding two recent text-to-speech synthesis networks. We include the standard and large BigVGAN (Lee et al., 2023a) architecture as well as the Avocodo (Bak et al., 2022) network. Finally, we employ integrated gradients (Sundararajan et al., 2017) to systematically explore the behavior of our models.

Project source code and the dataset extension are available online [1].

---

[1] https://github.com/gan-police/audiodeepfake-detection, https://zenodo.org/records/10512541

## 2 Related work

### 2.1 Generative Models

The MelGAN architecture (Kumar et al., 2019) was an early Generative Adversarial Network (GAN) in the audio domain. It proposed to work with mel-scaled spectrograms as an intermediate representation. The evolution of generative models with intermediate mel-representations continues with the HiFiGAN architecture (Kong et al., 2020). Its generator contains multiple residual blocks and its training procedure minimizes the L1 distance between ground truth and generated mel-spectrograms. Parallel WaveGAN (Yamamoto et al., 2020) integrates the WaveNet (Oord et al., 2016) architecture and uses the STFT intermediate representation. Similarly, WaveGlow (Prenger et al., 2019), combines a WaveNet backbone with a flow-based Glow paradigm (Kingma & Dhariwal, 2018). The aforementioned architectures are part of the WaveFake dataset (Frank & Schönherr, 2021), which we will study in detail.

Furthermore, novel Text to Speech (TTS) systems have appeared since the publication of the WaveFake dataset. Lee et al. (2023a), for example, trained the biggest vocoder to date. Additionally, their architecture shifts to periodic activation functions. The authors report excellent generalization properties. Further, the parallelly developed Avocodo network (Bak et al., 2022) aims to reduce artifacts by removing low-frequency bias. We additionally include both architectures in our study.

### 2.2 Audiofake detection

The ability of generative machine learning to generate credible media samples led to an investigation into their automatic detection. Wang et al. (2020a), for example, devised generative content detectors for images and found that Convolutional Neural Network (CNN)-detectors initialized on ImageNet do allow the detection of many other CNN-based image generators, even if trained on only a single image-generator. While the deep learning community has focused its attention on the image domain (Wang et al., 2020a; Wolter et al., 2022; Huang et al., 2022; Dong et al., 2022; Li et al., 2022; Frank et al., 2020; Frank & Schönherr, 2021; Schwarz et al., 2021; Dzanic et al., 2020), audio-generation has largely been neglected so far (Frank & Schönherr, 2021).

In the audio domain, Frank & Schönherr (2021) established a baseline by introducing the WaveFake dataset. The dataset includes five different generative network architectures in nine sample sets. In addition to collecting the data, two baseline models are established. Some related work studies the ASVspoof 2019 (Lavrentyeva et al., 2019) and 2021 (Tomilov et al., 2021) challenges. Müller et al. (2022), for example, evaluate multiple randomly initialized architectures. Their experiments include a ResNet18, a transformer, and the Light Convolutional Neural Network (LCNN) architecture proposed by Lavrentyeva et al. (2017). The LCNN-architecture introduces max feature maps, which split the channel dimension in two. The resulting map contains the elementwise maximum of both halves. Jung et al. (2020) propose to detect synthetic media samples using a similar architecture that combines a CNN with a gated recurrent unit. The authors call their model RawNet2. RawNet2 works directly on raw waveforms. The model did not generalize well on the original WaveFake dataset (Frank & Schönherr, 2021).

Furthermore, Zhang et al. (2017) make use of dilated convolutional networks for environmental sound classification. While CNNs often use small filters, limiting contextual information, dilated filters appear to represent contextual information without resolution loss (Yu & Koltun, 2015). Attention is another popular way to model long-term dependencies. Starting from Mel-spectrogram features Gong et al. (2021) propose the Audio Spectrogram Transformer (AST), a purely attention-based network. The authors observe state-of-the-art performance on audio classification tasks, which merits inclusion in this study.

### 2.3 Fingerprints of generative methods

Marra et al. (2019) propose to compute fingerprints of image generators by looking at noise residuals. Residuals are computed via $\mathbf{r}_i = \mathbf{x}_i - f(\mathbf{x}_i)$, where $i$ loops over the inputs. In their equation, $\mathbf{x}$ denotes inputs, and $f(\cdot)$ is a suitable denoising filter. In other words, Marra et al. (2019) obtain the residual by subtracting the low-level image content $f(\mathbf{x}_i)$. Assuming residuals contain a fingerprint $\mathbf{f}$ and a random

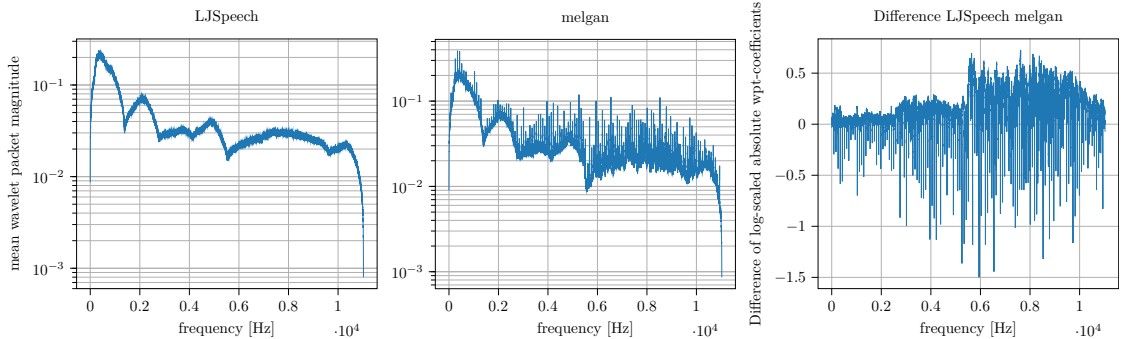

Figure 1: Mean level 14 Haar-Wavelet decomposition of original LJSpeech (left) recordings as well as synthetic versions generated by MelGAN (center). The difference between both plots is shown on the right. MelGAN (Kumar et al., 2019) displays a characteristic spike-shaped fingerprint. MelGAN produces characteristic spikes in the frequency domain.

noise-component $\mathbf{n}$, that is $\mathbf{r}_i = \mathbf{f} + \mathbf{n}_i$. The fingerprint is estimated via $\hat{\mathbf{f}} = \frac{1}{N}\sum_{i=1}^{N} \mathbf{r}_i$. Wang et al. (2020a) continued along this line of work and computed additional image generator fingerprints by averaging high-pass filtered Fourier spectra. By averaging enough samples from a generator, variable parts of each signal are lost, and the stable fingerprint remains. In the audio domain, Frank & Schönherr (2021) previously followed a similar logic and computed average frequency energies based on Fourier features.

### 2.4 Frequency analysis in audio processing

Frequency representations have a rich history in the field of audio processing. These representations are biologically motivated since the cochlea inside the human inner ear acts as a spectrum analyzer (Huang et al., 2001). Most of the literature chooses to work with the Short-Time Fourier Transform (STFT) (Tomilov et al., 2021; Palanisamy et al., 2020), or the Discrete Cosine Transform (DCT) (Sahidullah et al., 2015; Frank & Schönherr, 2021). After mapping the data to the frequency domain, the dimensionality is often reduced via a set of filter banks (Sahidullah et al., 2015). These can be linearly- or mel-spaced. Mel-spacing produces a high resolution in lower frequency ranges, where humans hear exceptionally well. Like mel-scaled STFT features, the Constant Q Transform (CQT) (Todisco et al., 2016) is perceptually motivated. The approach delivers a higher frequency resolution for lower frequencies and a higher temporal resolution for higher frequencies (Todisco et al., 2016). The process is similar to the fast wavelet transform. Wavelets have a long track record in engineering and signal processing. More recently, wavelet methods have started to appear in the artificial neural networks literature in the form of scatter-nets (Mallat, 2012; Cotter, 2020) and synthetic image detection (Wolter et al., 2022; Huang et al., 2022; Li et al., 2022). In the audio domain (Fathan et al., 2022) work with Mel-spectrogram inputs and feed features from a standard wavelet tree in parallel to a traditional CNN at multiple scales.

## 3 Experiments

To better understand how synthetic media differs from actual recordings, this section looks for artifacts that neural speech generators leave behind. We follow prior work and study Fourier-based representations. Additionally, we employ the Wavelet Packet Transform (WPT). This section recreates the experimental setup described by Frank & Schönherr (2021). We do so to allow easy comparisons. Frank & Schönherr (2021) work with the LJSpeech and JSUT data sets. Various generators are tasked to recreate LJSpeech or JSUT sentences, which allows direct comparison. We find artifacts across the entire spectrum. Consequently, we do not apply the high-pass filter step proposed by Marra et al. (2019) for images. Instead, we identify stable patterns all recordings have in common by averaging wavelet packet coefficients for 2500 single-second recordings per audio source. Artifact plots in this paper show log-scaled mean absolute wavelet-packet or Fourier coefficients up to 11.025 kHz. All spectra are averages over 2.500 recordings.

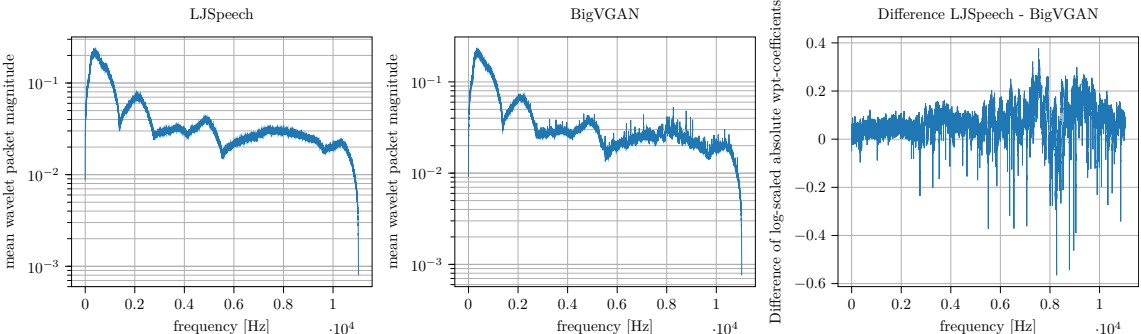

Figure 2: Mean level 14 Haar-Wavelet decomposition of original LJSpeech (left) recordings as well as synthetic versions generated by BigVGAN (center). The plot on the right shows the difference. The more recent BigVGAN produces spikes in the spectral representation. Albeit less pronounced than those of MelGAN, BigVGAN's spectral representation diverges from the original LJSpeech coefficients, especially for higher frequencies.

Figure 1 studies stable frequency domain patterns created by the MelGAN architecture. The figure shows mean absolute level 14 Haar-Wavelet packet transform coefficients for LJSpeech and MelGAN (Kumar et al., 2019) audio files. The transform reveals that MelGAN produces a spike-shaped pattern in the frequency domain. Supplementary Figure 16 confirms the spike pattern using a Fourier transform-based approach. We repeat the same analysis for the recent BigVGAN-architecture (Lee et al., 2023a). Figure 2 presents our results. Small spikes are visible in the center figure. The pattern resembles what we saw for MelGAN.

Supplementary section 6 includes Wavelet packet and Fourier Artifact visualizations for all generative networks in this study. Furthermore, we create an audible version of the mean spectra, by transforming it back into the time domain. The supplementary material contains sound files with amplified generator artifacts. The files are exciting but not aesthetically pleasing. We recommend listening at low volumes.

### 3.1 Training general detectors for synthetic audio

If similar artifacts appear for all generators, we should be able to design detectors for synthetic media from unknown sources. Our detectors must be able to detect synthetic audio from generators, which had not been part of the training set. We have reason to believe this is possible for images (Wang et al., 2020b). Yet for audio Frank & Schönherr (2021) reported that standard deep detectors had low recognition rates for samples from unknown generators. To solve this problem, we devise a new Dilated Convolutional Neural Network (DCNN) architecture.

We ran our experiments on a four compute-node cluster with two AMD EPYC 7402 2.8 GHz host CPUs and four NVidia A100 Tensor Core graphics cards per host with 40 GB memory each. All experiments require four GPUs. Our experimental work builds upon Pytorch (Paszke et al., 2019) and the Pytorch-Wavelet-Toolbox (Moritz Wolter, 2021). This experimental section studies the generalizing properties of deep fake-audio classifiers trained only on a single generator. We consider the WaveFake dataset (Frank & Schönherr, 2021) with all its generators, including the conformer-TTS samples and the Japanese language examples from the JSUT dataset.

We extend the WaveFake dataset by adding samples drawn from the Avocodo (Bak et al., 2022) and BigV-GAN (Lee et al., 2023a) architectures. Due to a lack of pre-trained weights, we retrained Avocodo using the publicly available implementation from (Bak et al., 2023) commit `2999557`. We trained for 346 epochs or 563528 steps. Hyperparameters were chosen according to Bak et al. (2022) with a learning rate of 0.0002 for the discriminator and generator. After training, we used their inference script to generate additional LJSpeech samples. Following a similar procedure, we add BigVGAN (Lee et al., 2023a) generated audio samples. We employ a BigVGAN Large (L) with 112 M parameters, and its downsized counterpart with 14 M

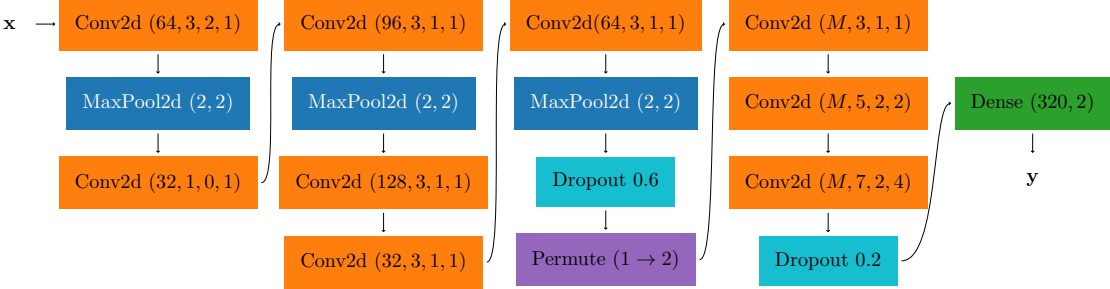

Figure 3: Structure of our Dilated Convolutional Neural Network (DCNN). The Conv2d blocks denote 2D-Convolution operations with hyperparameters (Output Channels, Kernel Size, Padding, Dilation). We always work with unit strides. Each Conv2d is preceded by a Batch Normalization Layer (Ioffe & Szegedy, 2015) and followed by a PreLU activation (Xu et al., 2015). The permutation operation permutes the first with the second dimension of the input (we consider the batch dimension to be dimension zero). $M$ denotes the number of output channels from the convolutional layers before.

weights, which we refer to as BigVGAN. We use the code from Lee et al. (2023b). Again, additional fake LJSpeech samples are generated with the authors' inference script for both models. The Japanese language (JSUT) samples from Frank & Schönherr (2021) are downsampled from 24 kHz to 22.05 kHz to ensure a uniform sampling rate for all recordings. All samples are cut into one-second segments. All sets contain an equal amount of real and fake samples.

### 3.1.1 Generalizing detectors for the extended WaveFake Dataset

Frank & Schönherr (2021) include MelGAN, Parallel WaveGAN, Multi-bad MelGAN, full-band MelGAN, HiFi-GAN, and WaveGlow in their Wavefake data set. We add samples from Avocodo and BigVGAN by re-synthesizing all utterances in the LJSpeech corpus, we obtain a total of 13 100 additional audio files for Avocodo, 13 100 extra files for BigVGAN, and another 13 100 for BigVGAN Large.

Frequency information was a crucial ingredient in previous work on images (Frank et al., 2020; Wolter et al., 2022). Consequently, we explore wavelet packets and the STFT as input representations. Both capture frequency information. Our input pipeline proceeds as follows: Audio is loaded first. After batching, we transform with either the WPT or STFT. Once the transformation is complete, we always take the absolute values of the result. Generally, we compute the square and add extra labels if we don't. Finally, the result is re-scaled using the natural logarithm (ln). The WPT depends on the underlying wavelet. We study the effect of wavelets from the Daubechies, Symlet, and Coiflet families.

According to Frank & Schönherr (2021), a Gaussian Mixture Model (GMM) trained on top of Linear Frequency Cepstral Coefficient (LFCC) features performed best on the original WaveFake dataset. The GMM outperformed the deep RawNet2 proposed by Jung et al. (2020). RawNet2 processes raw and unmodified waveforms. A convolution-based encoder computes feature vectors. After the encoder, RawNet2 employs a recurrent layer to integrate information over time. Instead of relying on recurrent connections, we process contextual information via dilated convolution. Dilated convolutions enlarge the receptive field without downsampling (Yu & Koltun, 2015). We found dilated convolutions delivered improved run-time and fake detection accuracy. Furthermore, our network employs the PReLU Xu et al. (2015) activation function, which performs well in tandem with dilated convolution Zhang et al. (2017). Figure 3 depicts our architecture.

Adam (Kingma & Ba, 2015) optimizes almost all networks with a learning rate of 0.0004. We follow Gong et al. (2021) and set the step size to 0.00004 for the AST. Each training step used 128 audio samples per batch. Finally, we employ weight decay and dropout. The L2 penalty is set to 0.001 unless stated otherwise. We retrain the LFCC-GMM using the same training setup as Frank & Schönherr (2021). LCNN training is based on our code. We adopt the AST implementation provided by Gong et al. (2021). We have to ensure generalization to unknown generators. Consequently, all detectors see *only* full-band MelGAN and real

Table 1: Fake detection results on the extended WaveFake dataset including JSUT. All input transforms work with 256-frequency bins. Our models are trained exclusively on samples drawn from a full-band MelGAN. We report the test set accuracy and aEER. To add additional context, we report mean test set accuracy and aEER and standard deviation over five runs for all experiments using seeds 0 to 4.

| Network | Input | Accuracy [%] | | aEER | |
|---|---|---|---|---|---|
| | | max | $\mu \pm \sigma$ | min | $\mu \pm \sigma$ |
| DCNN (ours) | STFT | 96.46 | $91.72 \pm 2.94$ | 0.036 | $0.159 \pm 0.150$ |
| | db4 | 95.31 | $90.84 \pm 5.59$ | 0.079 | $0.133 \pm 0.048$ |
| | db5 | 96.88 | $94.65 \pm 1.85$ | 0.048 | $0.082 \pm 0.042$ |
| | db8 | 98.23 | $89.11 \pm 8.80$ | 0.097 | $0.132 \pm 0.044$ |
| | sym4 | 97.73 | $94.31 \pm 3.05$ | 0.059 | $0.203 \pm 0.125$ |
| | sym5 | 97.70 | $95.25 \pm 3.09$ | 0.031 | $\mathbf{0.069 \pm 0.036}$ |
| | sym8 | 98.49 | $96.77 \pm 2.40$ | 0.062 | $0.145 \pm 0.078$ |
| | coif4 | 97.90 | $91.32 \pm 5.41$ | $\mathbf{0.025}$ | $0.084 \pm 0.047$ |
| | coif8 | 98.72 | $97.39 \pm 1.80$ | 0.026 | $0.079 \pm 0.047$ |
| LCNN | STFT | 91.65 | $79.21 \pm 16.55$ | 0.083 | $0.169 \pm 0.101$ |
| | sym5 | 97.46 | $90.12 \pm 6.44$ | 0.067 | $0.108 \pm 0.042$ |
| AST | STFT | 90.98 | $87.10 \pm 2.54$ | 0.089 | $0.122 \pm 0.021$ |
| | sym5 | 93.49 | $91.25 \pm 1.38$ | 0.065 | $0.087 \pm 0.013$ |
| GMM | LFCC (Frank & Schönherr) | – | – | 0.145 | – |

samples during training. Frank & Schönherr (2021) trained their best performing detectors exclusively on Full-band MelGAN, which motivates our choice. Test accuracies and average Equal Error Rates (aEERs) are computed for test samples from all eight generators and the original audio, where we measure our detector's ability to separate real and fake.

Table 1 lists our results for the extended WaveFake dataset. We find that deep neural networks do generalize well to unseen generators. LCNN, DCNN and AST outperform the GMM proposed by Frank & Schönherr (2021). Furthermore, we observe improvements for our DCNN compared to the LCNN and the AST. In addition to the accuracy improvements, we reduce the number of optimizable parameters. For STFT and sym5-WPT inputs the DCNN produces better results on average and when considering only the best run. An LCNN has 3,312,450 parameters. Our DCNN uses only 239,015 for a sym5 input. Which leads to an additional efficiency improvement. A complete list of model parameter numbers is available in supplementary Table 9. Regarding the choice of the wavelet, we observe the average performance for the sym5 wavelet, while the coif4 input WPT leads to the best-performing network. Supplementary Table 6 lists results for all wavelets we tested.

### 3.1.2 The original WaveFake dataset

We continue to study the original WaveFake (Frank & Schönherr, 2021) dataset in isolation. Table 2 enumerates the performance of our classifiers on this unmodified dataset. We find a different picture in comparison to Table 1. Excluding the newer BigVGAN and Avocodo networks shifts the observed performance in favor of networks trained on STFT inputs. Please consider Table 3 to understand the root of the differences we previously observed in section 3.1.2. Table 3 reveals the performance of our classifiers when tasked to identify our most recent vocoders in isolation. Our STFT based networks struggle with identifying BigVGAN samples. The WPT based networks do much better on this task, which explains the differences between Tables 1 and 2. Further, in comparison to our DCNN, the GMM from Frank & Schönherr (2021) drops in performance when evaluated on the newer generators. Similarly AST performs well on the original Wave-Fake data but struggles to identify BigVGAN samples. In summary, the DCNN improves upon the numbers presented by Frank & Schönherr (2021) for both coif4 and STFT inputs.

Table 2: Results on the unmodified WaveFake dataset (Frank & Schönherr, 2021). We cite baseline numbers as reported in the original paper. All frequency representations work with 256 bins. Our network outperforms the GMM-LFCC and their RawNet2 implementation for the db4, db5, sym4, sym5, coif4 as well as the coif8 wavelet, and the STFT. The AST produces the best average performance over multiple seeds on this smaller dataset.

| Network | Input | Accuracy [%] | | aEER | |
|---|---|---|---|---|---|
| | | max | $\mu \pm \sigma$ | min | $\mu \pm \sigma$ |
| DCNN (ours) | STFT | 99.88 | $97.98 \pm 3.18$ | **0.001** | $0.099 \pm 0.178$ |
| | db4 | 94.71 | $90.20 \pm 5.90$ | 0.079 | $0.136 \pm 0.050$ |
| | db5 | 96.36 | $94.39 \pm 1.45$ | 0.048 | $0.083 \pm 0.041$ |
| | db8 | 98.21 | $89.17 \pm 8.83$ | 0.100 | $0.133 \pm 0.046$ |
| | sym4 | 97.24 | $94.09 \pm 2.26$ | 0.057 | $0.200 \pm 0.127$ |
| | sym5 | 97.60 | $95.57 \pm 2.58$ | 0.032 | $0.066 \pm 0.035$ |
| | sym8 | 98.80 | $96.92 \pm 1.69$ | 0.062 | $0.142 \pm 0.081$ |
| | coif4 | 98.24 | $92.88 \pm 5.05$ | 0.025 | $0.118 \pm 0.088$ |
| | coif8 | 98.81 | $97.87 \pm 0.92$ | 0.026 | $0.121 \pm 0.089$ |
| LCNN | STFT | 99.88 | $98.33 \pm 1.85$ | **0.001** | $0.019 \pm 0.018$ |
| | sym5 | 96.89 | $95.34 \pm 1.83$ | 0.037 | $0.085 \pm 0.053$ |
| AST | STFT | 99.37 | $98.31 \pm 1.49$ | 0.007 | **0.018 ± 0.016** |
| | sym5 | 93.63 | $91.98 \pm 0.98$ | 0.065 | $0.081 \pm 0.010$ |
| GMM | LFCC (Frank & Schönherr) | – | – | 0.062 | – |
| RawNet2 | raw (Frank & Schönherr) | – | – | 0.363 | – |

Table 3: Results on BigVGAN (Lee et al., 2023a) and Avocodo (Bak et al., 2022). When looking at the two newer generators in isolation, we observe aEERs in line with previous results for our wavelet-based classifiers. The GMM-LFCC (Frank & Schönherr, 2021) and the networks with STFT-inputs drop in performance when evaluated on the unseen BigVGAN.

| Network | Input | Avocodo | | | | BigVGAN (L) | | | |
|---|---|---|---|---|---|---|---|---|---|
| | | Accuracy [%] | | aEER | | Accuracy [%] | | aEER | |
| | | max | $\mu \pm \sigma$ | min | $\mu \pm \sigma$ | max | $\mu \pm \sigma$ | min | $\mu \pm \sigma$ |
| DCNN (ours) | STFT | 99.99 | $91.40 \pm 16.49$ | **0.001** | $0.096 \pm 0.179$ | 84.67 | $65.12 \pm 11.82$ | 0.182 | $0.365 \pm 0.095$ |
| | db4 | 98.46 | $94.11 \pm 3.69$ | 0.024 | $0.098 \pm 0.056$ | 93.78 | $89.68 \pm 2.92$ | 0.079 | $0.136 \pm 0.047$ |
| | db5 | 98.60 | $96.13 \pm 2.87$ | 0.026 | $0.064 \pm 0.048$ | 97.47 | $93.31 \pm 3.09$ | 0.052 | **0.097 ± 0.047** |
| | db8 | 99.53 | $96.92 \pm 2.23$ | 0.005 | $0.055 \pm 0.041$ | 94.88 | $87.77 \pm 6.81$ | 0.101 | $0.159 \pm 0.061$ |
| | sym4 | 97.85 | $86.58 \pm 11.73$ | 0.033 | $0.184 \pm 0.142$ | 93.58 | $86.55 \pm 5.57$ | 0.094 | $0.226 \pm 0.108$ |
| | sym5 | 99.34 | $97.48 \pm 2.21$ | 0.010 | $0.046 \pm 0.039$ | 96.28 | $92.66 \pm 4.02$ | **0.043** | **0.097 ± 0.051** |
| | sym8 | 99.76 | $91.82 \pm 6.43$ | 0.003 | $0.130 \pm 0.095$ | 95.08 | $90.85 \pm 3.69$ | 0.064 | $0.166 \pm 0.071$ |
| | coif4 | 99.72 | $95.24 \pm 7.07$ | 0.005 | $0.072 \pm 0.101$ | 95.94 | $87.23 \pm 7.40$ | 0.047 | $0.173 \pm 0.093$ |
| | coif8 | 98.70 | $92.31 \pm 6.58$ | 0.025 | $0.122 \pm 0.090$ | 96.11 | $93.46 \pm 3.32$ | 0.058 | $0.128 \pm 0.083$ |
| LCNN | STFT | 99.89 | $99.67 \pm 0.19$ | **0.001** | **0.006 ± 0.004** | 70.32 | $57.71 \pm 10.42$ | 0.308 | $0.382 \pm 0.060$ |
| | sym5 | 98.91 | $95.81 \pm 3.66$ | 0.011 | $0.069 \pm 0.060$ | 93.76 | $92.05 \pm 1.45$ | 0.069 | $0.112 \pm 0.042$ |
| AST | STFT | 99.61 | $98.84 \pm 0.62$ | 0.005 | $0.021 \pm 0.012$ | 62.95 | $50.41 \pm 7.48$ | 0.357 | $0.424 \pm 0.039$ |
| | sym5 | 98.18 | $97.86 \pm 0.28$ | 0.023 | $0.028 \pm 0.004$ | 91.99 | $88.53 \pm 2.36$ | 0.095 | $0.139 \pm 0.029$ |
| GMM | LFCC | – | – | 0.316 | – | – | – | 0.432 | – |

Table 4: DCNN fake-detection results on the extended WaveFake dataset (with JSUT) comparing a modified version of our architecture depicted in figure 3 with small modifications to motivate the use of max pooling and dropout. All input transforms work with 256-frequency bins. Our models are trained exclusively on samples drawn from a full-band MelGAN. We report the test set accuracy and aEER. To add additional context, we report mean test set accuracy and aEER and standard deviation over five runs for all experiments using seeds 0 to 4.

| DCNN-variant | Input | Accuracy [%] | | aEER | |
|---|---|---|---|---|---|
| | | max | $\mu \pm \sigma$ | min | $\mu \pm \sigma$ |
| | sym5 | 97.70 | $95.25 \pm 3.09$ | 0.031 | $0.069 \pm 0.036$ |
| | coif4 | 97.90 | $91.32 \pm 5.41$ | **0.025** | $0.084 \pm 0.047$ |
| without Max Pooling | sym5 | 97.39 | $95.59 \pm 1.73$ | 0.056 | $0.082 \pm 0.029$ |
| without Dropout | sym5 | 97.75 | $95.74 \pm 1.41$ | 0.049 | $\mathbf{0.062 \pm 0.014}$ |
| without Dropout | coif4 | 97.60 | $93.25 \pm 5.28$ | 0.029 | $0.073 \pm 0.044$ |
| without Dilation | sym5 | 93.02 | $87.76 \pm 3.21$ | 0.070 | $0.117 \pm 0.028$ |
| without Dilation | coif4 | 98.46 | $81.16 \pm 14.96$ | 0.040 | $0.173 \pm 0.094$ |

Table 5: The effect of the sign on the extended WaveFake dataset. We additionally concatenate a tensor with the sign patterns onto each input and retrain our networks.

| Input | Accuracy [%] | | aEER | |
|---|---|---|---|---|
| | max | $\mu \pm \sigma$ | min | $\mu \pm \sigma$ |
| SG-db5 | 87.69 | $83.03 \pm 3.14$ | 0.122 | $0.156 \pm 0.024$ |
| SG-db8 | 91.31 | $85.43 \pm 5.96$ | 0.086 | $0.134 \pm 0.045$ |
| SG-sym5 | 92.76 | $74.86 \pm 13.47$ | 0.073 | $0.203 \pm 0.096$ |
| SG-sym8 | 94.64 | $77.77 \pm 15.19$ | 0.054 | $0.180 \pm 0.105$ |
| SG-coif8 | 91.18 | $84.17 \pm 5.40$ | 0.087 | $0.147 \pm 0.041$ |

# 4   Ablation Study

This section ablates network parts and questions our design choices. We start with the network architecture. We either leave out all max pooling layers in exchange for a stride two downsampling in the layer before or leave out the Dropout layer. Table 4 lists the accuracy and aEER values we measured. The proposed DCNN architecture is robust. It works without max-pooling or dropout. However, the best individual networks employed both, which justifies their use in our experiments. We additionally ablate the effect of dilation. Table 4 shows significant performance drops in this case. We believe this finding confirms our initial intuition, which led us to replace the recurrent connections, which Jung et al. (2020) relied on with dilation.

## 4.1   Ablating the effect of the sign

The sign pattern is important for the WPT. Assume a wavelet packet tensor $\mathbf{P}$, which is obtained by stacking the end nodes of the packet graph. Given the pattern of minus signs $\mathbf{N}$, we could invert the rescaling since $\exp(\ln(\mathrm{abs}(\mathbf{P}))) \odot \mathbf{N} = \mathbf{P}$, if $\odot$ denotes the element wise Hadamard-product. Conserving the signs avoids information loss. For the WPT, we additionally explore two-channel inputs, where we tag on the sign pattern of the WPT. We call this case signed (SG). We argue that a lossless input is essential from a theoretical point of view. We observe no experimental benefit in Table 5. Consequently, we do not consider the sign pattern elsewhere in this study.

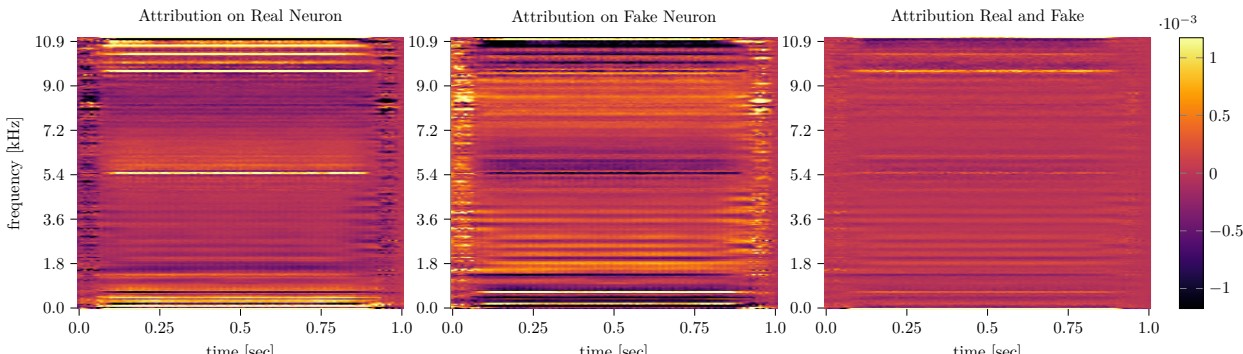

Figure 4: Attribution using integrated gradients on the-sym8-WPT-based DCNN-classifier evaluated over 2500 real audio samples from LJSpeech (left), 2500 fake audio samples from our extended version of the WaveFake dataset (middle) and averaged over both real and fake audios (right). We observe high values in the high-frequency domain as well as an inverse character between real and fake audio files. This WPT based CNN shows distinguishable interest regions for the model in several frequency bins.

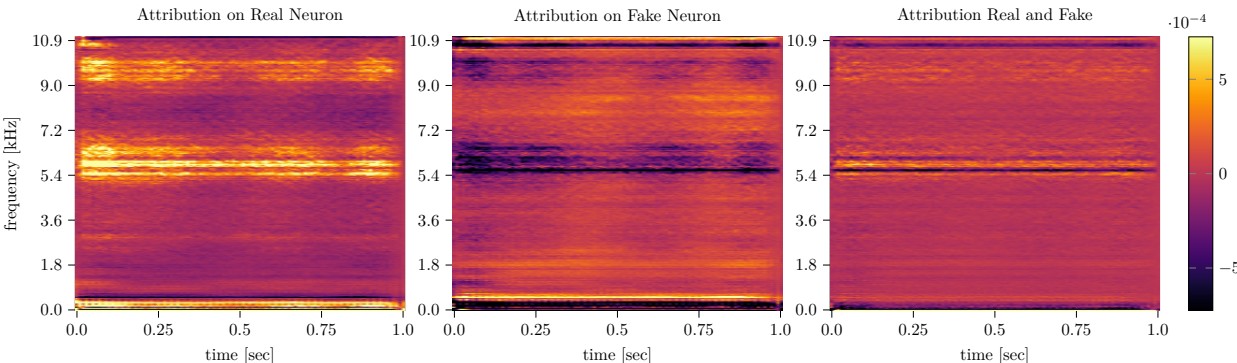

Figure 5: Attribution using integrated gradients (Sundararajan et al., 2017) on the STFT-DCNN-classifier for 2500 real audio samples from LJSpeech (left), 2500 fake audio samples from our extended version of the WaveFake dataset (middle) and averaged over both real and fake audios (right). We observe moderately high values in the high-frequency domain as well as an inverse character between real and fake audios similar to what we saw in Figure 4.

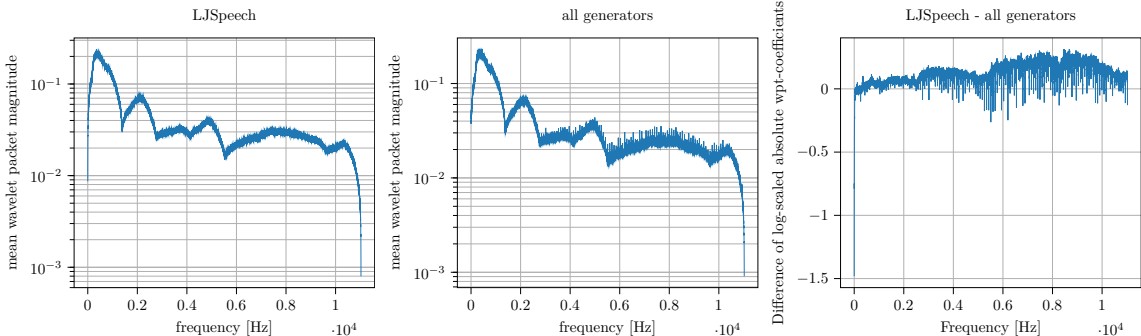

Figure 6: Mean level 14 Haar-Wavelet decomposition of original LJSpeech (left) recordings as well as synthetic versions generated by all generators in the extended WaveFake dataset Frank & Schönherr (2021) (center). The difference between both plots is shown on the right.

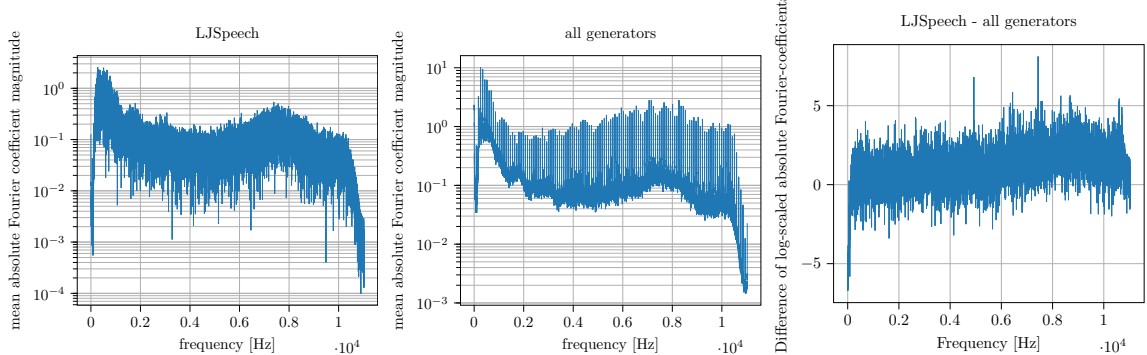

Figure 7: Mean absolute Fourier coefficient visualizations of original LJSpeech (left) recordings as well as synthetic versions generated by all generators in the extended WaveFake dataset Frank & Schönherr (2021) (center). The difference between both plots is shown on the right.

## 4.2 Attribution with Integrated Gradients

Figure 4 illustrates attribution using integrated gradients as proposed by Sundararajan et al. (2017) on the sym8-WPT input-network evaluated on the extended WaveFake test-dataset. We distinguish between the regions of interest within the input, averaged across 2500 samples of real audio samples (left plot) and 2500 samples of fake audio samples (middle), in order to identify areas of significance for the artificial neural network. This distinction is also made by comparing the averaged attribution across fake and real audio samples (right plot). Similarly, we repeat the same experiment for the STFT version of the DCNN. Figure 5 plots the result. We observe that, for both transformations, the high-frequency components play a significant role in the classification process. Both transformations exhibit an inverse character in terms of attribution between real and fake audio samples. In Figures 1 and 2, we saw differences across the spectrum, while higher frequencies tended to differ more. Figures 4 and 5 reflect this pattern. Generally, all frequencies impact the result, but frequencies above 4kHz matter more. Figures 6 and 7 allow us to sanity check the integrated gradient plots. The integrated gradient indicated impoact across the spectrum, with higher influence at higher frequencies. The mean spectra differ more at higher frequencies, which validates our previous observation.

## 5   Conclusion

This paper investigated the WaveFake dataset and its extended version with two additional recent GAN models. Frank & Schönherr (2021) observed poor generalization for deep neural network-based fake detectors. In response, the authors proposed using traditional methods to mitigate overfitting to specific audio generators. We found stable patterns in the averaged spectra for all generators. Building upon this observation, DCNN-based classifier demonstrated robust generalization performance across various representations in the two-dimensional frequency domain. This observation remained consistent regardless of the chosen representation method. Moreover, our attribution analysis highlighted the significance of high-frequency components. It revealed distinct attribution patterns between real and fake audio samples, underscoring the interpretability and potential insights provided by the investigated transformations.

### Acknowledgments

Research was supported by the Bundesministerium für Bildung und Forschung (BMBF) via the WestAI and BnTrAInee projects. The authors gratefully acknowledge the Gauss Centre for Supercomputing e.V. (www.gauss-centre.eu) for funding this project by providing computing time through the John von Neumann Institute for Computing (NIC) on the GCS Supercomputer JUWELS at Jülich Supercomputing Centre (JSC). Last but not least we would like to thank our anonymous reviewers for their feedback, which greatly improved this paper.

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

## 6 Supplementary

## Acronyms

**aEER** average Equal Error Rate

**AST** Audio Spectrogram Transformer

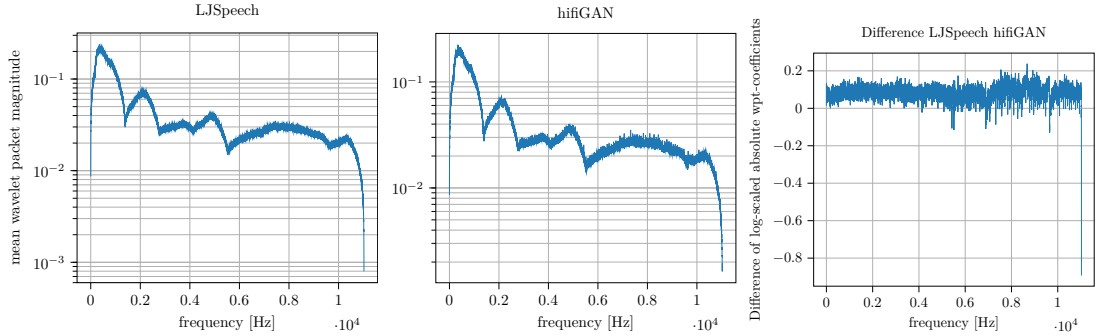

Figure 8: Mean level 14 Haar-Wavelet coefficients of LJSpeech (left) recordings as well as versions generated by the Hifi-GAN-architecture (Kong et al., 2020). We see some spikes for the Hifi-GAN-architecture (Kong et al., 2020) as well, but MelGAN produces more prominent spikes. The difference plot reveals that HifiGAN produces smaller coefficients on average than the true distribution.

**CNN** Convolutional Neural Network

**CQT** Constant Q Transform

**DCNN** Dilated Convolutional Neural Network

**DCT** Discrete Cosine Transform

**FFT** Fast Fourier Transform

**FWT** Fast Wavelet Transform

**GAN** Generative Adversarial Network

**GMM** Gaussian Mixture Model

**LCNN** Light Convolutional Neural Network

**LFCC** Linear Frequency Cepstral Coefficient

**ln** natural logarithm

**STFT** Short-Time Fourier Transform

**TTS** Text to Speech

**WPT** Wavelet Packet Transform

## 6.1 Additional Fourier-Artifact plots

Fourier Transform based energy difference plots previously also appeared in Frank & Schönherr (2021). We confirm their results and show Fourier-Transform-based plots again to allow easy comparison with the wavelet packet approach.

## 6.2 The Fast Wavelet Transform (FWT)

The FWT works with two filter pairs. An analysis pair $\mathbf{h}_\mathcal{L}$, $\mathbf{h}_\mathcal{H}$ and a synthesis pair $\mathbf{f}_\mathcal{L}$, $\mathbf{f}_\mathcal{H}$. The analysis transform relies on convolutions with a stride of two. The first level of the transform computes

$$\mathbf{f}_\mathcal{L} *_{\downarrow 2} \mathbf{x} = \sum_k \mathbf{h}_\mathcal{L}[k]\mathbf{x}[n-k] = \mathbf{y}_a[n], \tag{1}$$

$$\mathbf{f}_\mathcal{H} *_{\downarrow 2} \mathbf{x} = \sum_k \mathbf{h}_\mathcal{H}[k]\mathbf{x}[n-k] = \mathbf{y}_d[n]. \tag{2}$$

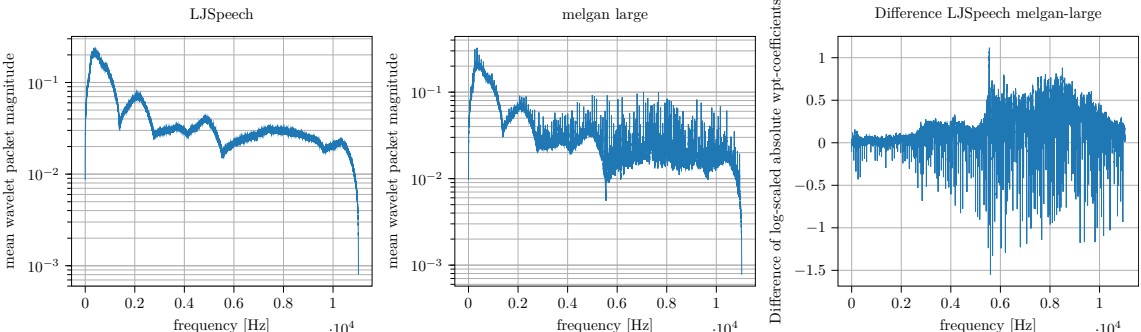

Figure 9: Mean absolute wavelet packet plots for the melgan-large and multi-band melgan generators alongside mean coefficients from the original data set. We see melgan-like spikes for the large melgan architecture.

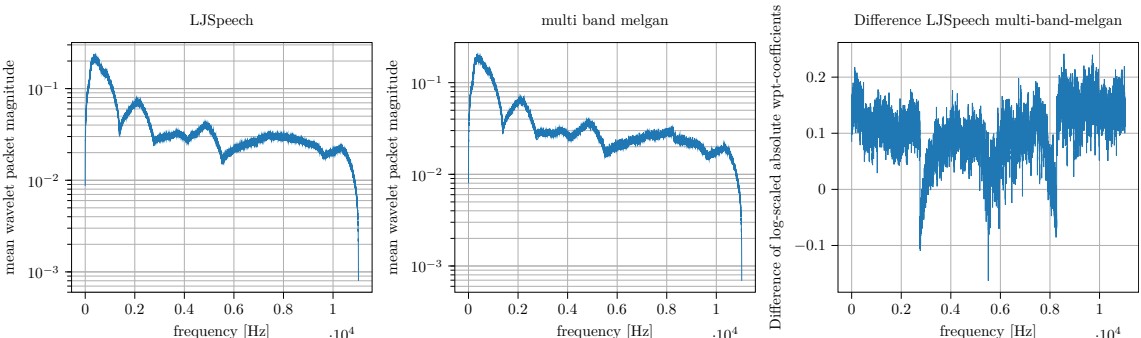

Figure 10: The multiband melgan's (Yang et al., 2021) mean absolute coefficients reveal a discontinuity between 8 and 10kHz.

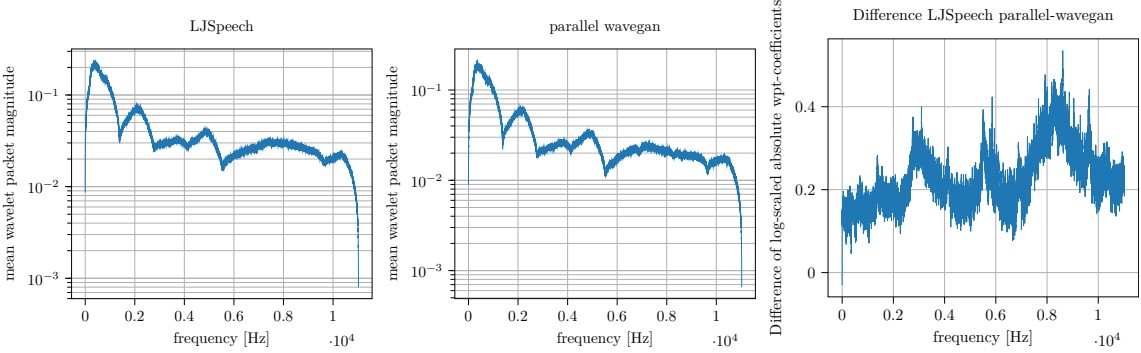

Figure 11: Parallel Wavegan (Yamamoto et al., 2020)

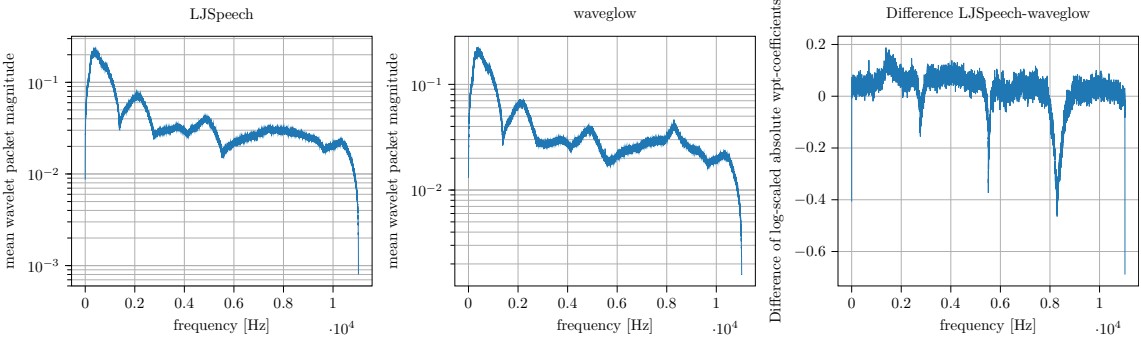

Figure 12: WaveGlow (Prenger et al., 2019)

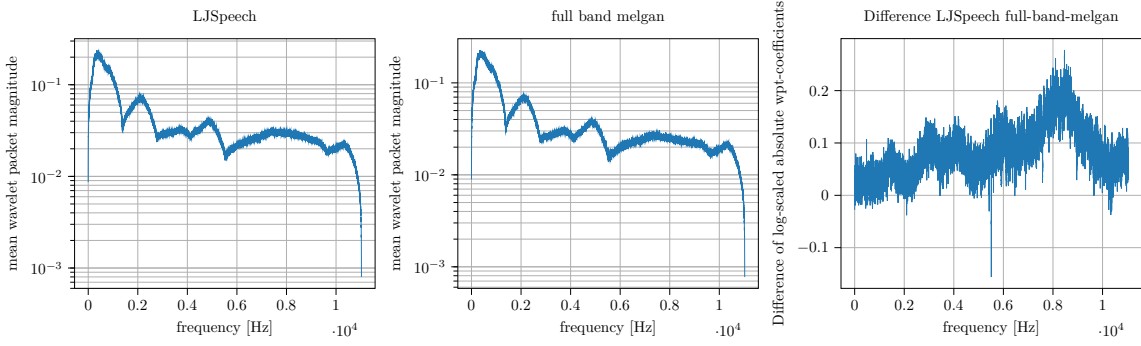

Figure 13: Full-band MelGAN plots.

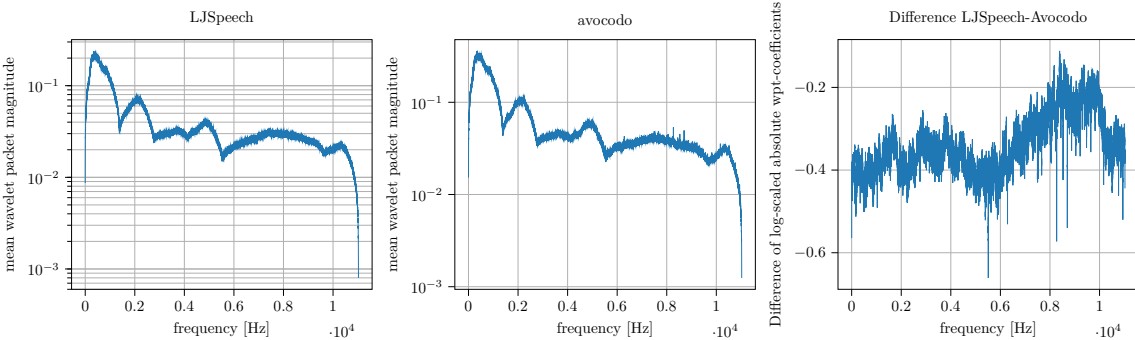

Figure 14: Avocodo plots.

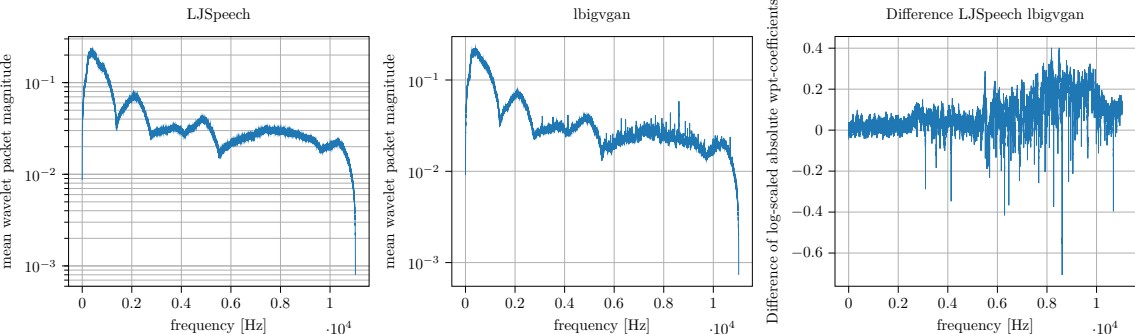

Figure 15: BigVGAN Large plots.

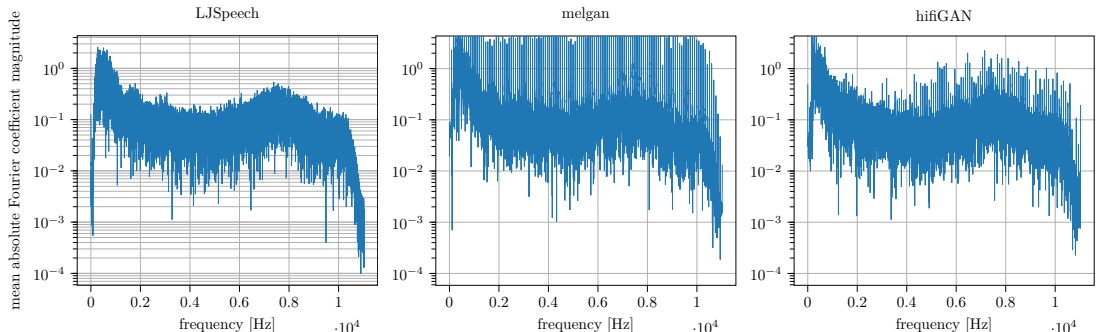

Figure 16: Mean absolute Fourier coefficient visualizations. Some vocoders leave distinct patterns in the frequency domain. Compared to the plot computed using the original LJSpeech recording in the top left, we see distinct spike patterns for the MelGAN and HiFiGAN vocoders. The real audio samples (left) come from the LJSpeech corpus. LJSpeech also served as a training dataset for both generators. Not all generators display patterns like these.

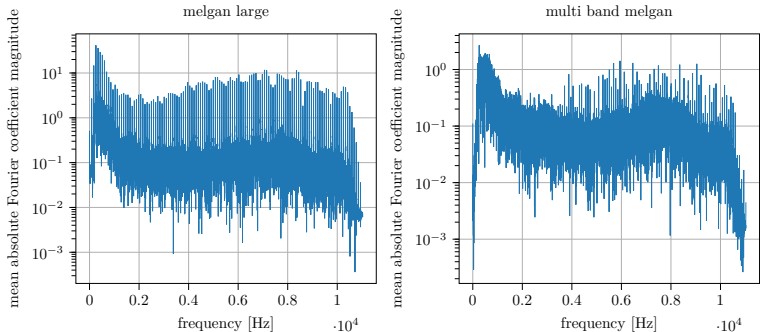

Figure 17: Fingerprint plots for MelGAN large and multi-band MelGAN generators. We observe clear spike-like patterns in the generated audio samples compared to the real LJSpeech samples.

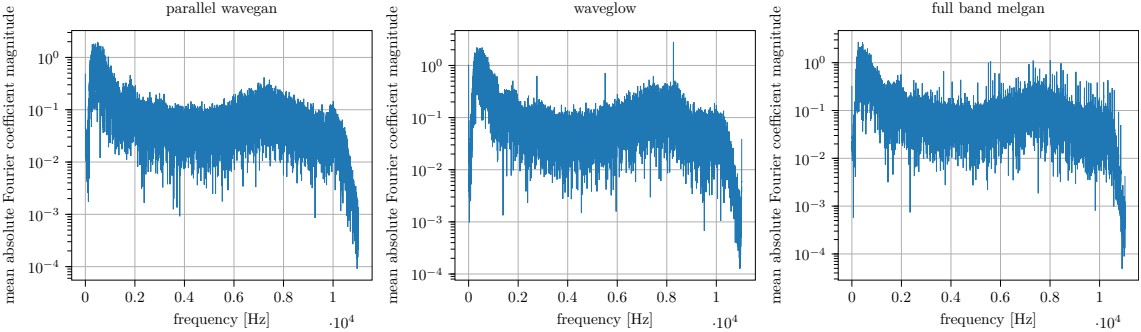

Figure 18: Fingerprint plots for the parallel WaveGAN, WaveGlow, and full-band MelGAN architectures. Especially for full-band MelGAN, we observe a clear spike-like pattern in the generated audio samples compared to the real LJSpeech samples.

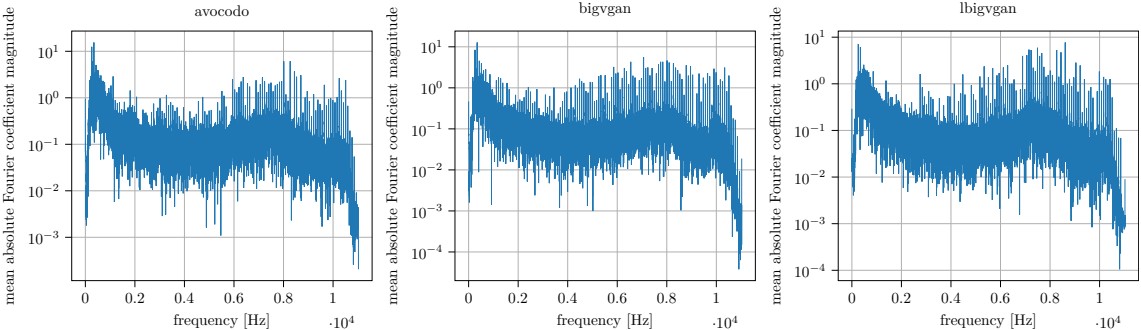

Figure 19: Fingerprint plots for the Avocodo, BigVGAN and Large BigVGAN architectures. Similarly to older generators, we observe spike-like patterns in the fingerprint plots of newer generators as well.

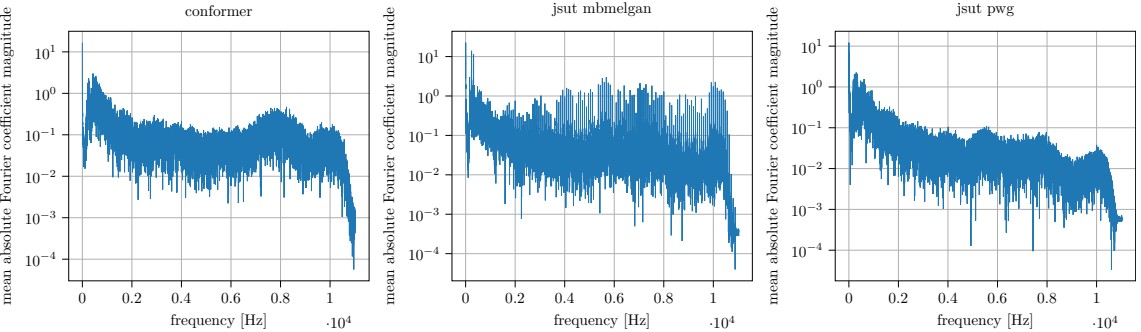

Figure 20: Fingerprint plots for the Conformer, JSUT multi-band MelGAN, and JSUT parallel WaveGAN architectures.

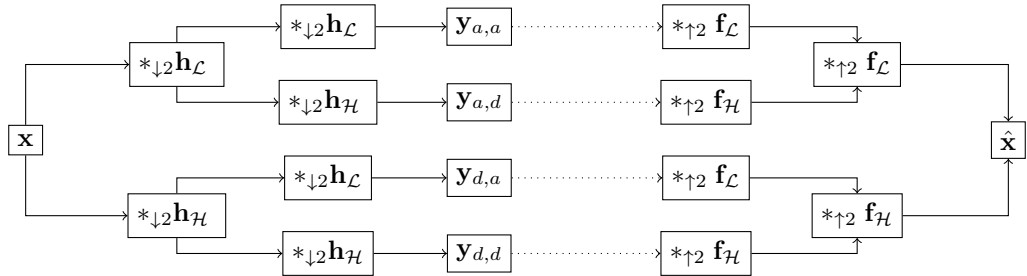

Figure 21: Visualisation of the wavelet packet transform. The figure uses **h** to denote analysis filters, **f** for synthesis filters, and stars $*$ for the convolution operation. The 0 subindex denotes lowpass- the 1 index denotes highpass-filters. $\downarrow 2$ denotes subsampling with a stride of two. $\uparrow 2$ denotes a transposed convolution with an upsampling effect and stride two. The notation follows Strang & Nguyen (1996). The **b**s denote the packet coefficients. The operation is invertible. Reconstruction produces an input-reconstruction $\hat{\mathbf{x}}$.

Index $k$ runs from 1 up to the filter length. Valid filter positions are indexed by $n$. $*_{\downarrow 2}$ symbolizes convolution with a stride of two. Padding is required to ensure every location is covered. Approximation coefficients have an $a$ subindex. Detail coefficients use $d$. For the next level we compute $\mathbf{y}_{aa} = \mathbf{f}_{\mathcal{L}} *_{\downarrow 2} \mathbf{y}_a$ and $\mathbf{y}_{ad} = \mathbf{f}_{\mathcal{H}} *_{\downarrow 2} \mathbf{y}_a$. The process continues by recursively filtering the approximation coefficients. Introducing convolution matrices **H** allows writing the analysis transform in matrix form Strang & Nguyen (1996),

$$\mathbf{A} = \dots \left( \begin{array}{c|c} \begin{matrix} \mathbf{H}_{\mathcal{L}} \\ \mathbf{H}_{\mathcal{H}} \end{matrix} & \\ \hline & \mathbf{I} \end{array} \right) \begin{pmatrix} \mathbf{H}_{\mathcal{L}} \\ \mathbf{H}_{\mathcal{H}} \end{pmatrix}. \tag{3}$$

the matrix structure illustrates how the process decomposes multiple scales. $\mathbf{H}_{\mathcal{L}}$ and $\mathbf{H}_{\mathcal{H}}$ denotes high-pass and low-pass convolution matrices, respectively. Evaluating the matrix-product yields **A**, the analysis matrix. In the factor matrices, the identity in the lower block grows with every scale. Therefore the process speeds up with every scale. The process is invertible. We could construct a synthesis matrix **S** using transposed convolution matrices to undo the transform. See Strang & Nguyen (1996) or Jensen & la Cour-Harbo (2001) for excellent in-depth discussions of the FWT.

## 6.3 Wavelet-packets

Related work (Frank et al., 2020; Wolter et al., 2022; Frank & Schönherr, 2021; Schwarz et al., 2021; Dzanic et al., 2020) highlights the importance of high-frequency bands for detecting gan-generated images. Consequently, expanding only the low-frequency part of the discrete wavelet-transform will not suffice. Instead, we employ the wavelet packet transform (Jensen & la Cour-Harbo, 2001). To ensure a fine-grained frequency resolution, we expand the complete tree. This approach is also known as the Walsh-form (Strang & Nguyen, 1996) of the transform. Figure 21 illustrates the idea. The main idea is to recursively expand the tree in the low- and high-pass direction. We index tree nodes with the filter sequences. $a$ denotes a low-pass or approximation, $d$ a high-pass or detail step. Via full expansion, we improve the resolution in the relevant high-frequency part of the tree compared to the standard FWT. The full tree linearly subdivides the spectrum from zero until the Nyquist frequency at half the sampling rate.

Not all filters are suitable choices for the analysis and synthesis pairs. Appropriate choices must obey the perfect reconstruction and anti-aliasing conditions Strang & Nguyen (1996). We found smaller Daubechies wavelets to be a good starting point. Symlets are a variant of the Daubechies-Wavelet family. Evaluating these symmetric cousins is usually the next step when exploring wavelet choices.

## 6.4 The Short-Time Fourier Transform (STFT)

By comparison, the STFT achieves localization in time via a sliding window. A sliding window **w** segments the input prior to Fast Fourier Transform (FFT). Only the part where the window is non-zero is visible. Or

Table 6: Complete source identification results on the extended WaveFake Frank & Schönherr (2021) dataset (with JSUT).

| | Method | Accuracy [%] | | aEER | |
|---|---|---|---|---|---|
| | | max | $\mu \pm \sigma$ | min | $\mu \pm \sigma$ |
| DCNN | STFT | 96.46 | $91.72 \pm 2.94$ | 0.036 | $0.159 \pm 0.150$ |
| | haar | 62.27 | - | 0.343 | - |
| | db2 | 93.06 | $74.38 \pm 11.01$ | 0.194 | $0.279 \pm 0.059$ |
| | db3 | 93.07 | $84.46 \pm 12.18$ | 0.098 | $0.159 \pm 0.072$ |
| | db4 | 95.31 | $90.84 \pm 5.59$ | 0.079 | $0.133 \pm 0.048$ |
| | db5 | 96.88 | $94.65 \pm 1.85$ | 0.048 | $0.082 \pm 0.042$ |
| | db6 | 97.01 | $89.58 \pm 10.95$ | 0.038 | $0.215 \pm 0.146$ |
| | db7 | 97.52 | $92.78 \pm 7.60$ | 0.040 | $0.112 \pm 0.081$ |
| | db8 | 98.23 | $89.11 \pm 8.80$ | 0.097 | $0.132 \pm 0.044$ |
| | db9 | 98.19 | $94.72 \pm 3.60$ | 0.050 | $0.095 \pm 0.034$ |
| | db10 | 98.08 | $90.56 \pm 8.15$ | 0.043 | $0.124 \pm 0.049$ |
| | sym2 | 93.06 | $74.38 \pm 11.01$ | 0.194 | $0.279 \pm 0.059$ |
| | sym3 | 93.07 | $84.46 \pm 12.18$ | 0.098 | $0.159 \pm 0.072$ |
| | sym4 | 97.73 | $94.31 \pm 3.05$ | 0.059 | $0.203 \pm 0.125$ |
| | sym5 | 97.70 | $95.25 \pm 3.09$ | 0.031 | $\mathbf{0.069 \pm 0.036}$ |
| | sym6 | 97.84 | $97.42 \pm 0.39$ | 0.048 | $0.172 \pm 0.094$ |
| | sym7 | 97.47 | $92.27 \pm 7.75$ | 0.067 | $0.172 \pm 0.089$ |
| | sym8 | 98.49 | $96.77 \pm 2.40$ | 0.062 | $0.145 \pm 0.078$ |
| | sym9 | 98.67 | $95.85 \pm 2.78$ | 0.060 | $0.112 \pm 0.064$ |
| | sym10 | 98.39 | $86.65 \pm 18.35$ | 0.028 | $0.148 \pm 0.113$ |
| | coif2 | 96.43 | $87.86 \pm 9.78$ | 0.100 | $0.180 \pm 0.089$ |
| | coif3 | 98.02 | $93.34 \pm 4.95$ | 0.042 | $0.085 \pm 0.036$ |
| | coif4 | 97.90 | $91.32 \pm 5.41$ | $\mathbf{0.025}$ | $0.084 \pm 0.047$ |
| | coif5 | 98.45 | $93.51 \pm 3.44$ | 0.057 | $0.221 \pm 0.175$ |
| | coif6 | 98.72 | $89.92 \pm 14.99$ | 0.035 | $0.115 \pm 0.095$ |
| | coif7 | 98.63 | $85.64 \pm 13.14$ | 0.045 | $0.128 \pm 0.097$ |
| | coif8 | 98.72 | $97.39 \pm 1.80$ | 0.026 | $0.079 \pm 0.047$ |
| | coif9 | 98.58 | $95.64 \pm 3.92$ | 0.051 | $0.106 \pm 0.044$ |
| | coif10 | 98.38 | $95.55 \pm 2.75$ | 0.039 | $0.067 \pm 0.025$ |
| LCNN | STFT | 91.65 | $79.21 \pm 16.55$ | 0.083 | $0.169 \pm 0.101$ |
| | sym5 | 97.46 | $90.12 \pm 6.44$ | 0.067 | $0.108 \pm 0.042$ |
| AST | STFT | 90.98 | $87.10 \pm 2.54$ | 0.089 | $0.122 \pm 0.021$ |
| | sym5 | 93.49 | $91.25 \pm 1.38$ | 0.065 | $0.087 \pm 0.013$ |
| GMM | LFCC (Frank & Schönherr) | – | – | 0.145 | – |

formally Griffin & Lim (1984)

$$\mathbf{X}[\omega, Sm] = \mathcal{F}(\mathbf{w}[Sm - l]\mathbf{x}[l]) = \sum_{l=-\infty}^{\infty} \mathbf{w}[Sm - l]\mathbf{x}[l]e^{-j\omega l}. \tag{4}$$

Here $\mathcal{F}$ denotes the classic discrete FFT. $S$ denotes the sampling period. We select specific windows with $m$, while $\omega$ denotes different frequencies in each window. In other words, after multiplying slices of $\mathbf{x}$ with $\mathbf{w}$, the result is Fourier-transformed.

Table 7: All our results on the original WaveFake dataset Frank & Schönherr (2021).

|  | Method | WaveFake Frank & Schönherr (2021) | | | |
|  |  | Accuracy [%] | | aEER | |
|  |  | max | $\mu \pm \sigma$ | min | $\mu \pm \sigma$ |
|---|---|---|---|---|---|
| DCNN | STFT | 99.88 | $97.98 \pm 3.18$ | **0.001** | $0.099 \pm 0.178$ |
|  | haar | 68.64 | - | 0.301 | - |
|  | db2 | 93.39 | $80.59 \pm 8.34$ | 0.132 | $0.237 \pm 0.076$ |
|  | db3 | 93.35 | $85.96 \pm 11.08$ | 0.084 | $0.152 \pm 0.070$ |
|  | db4 | 94.71 | $90.20 \pm 5.90$ | 0.079 | $0.136 \pm 0.050$ |
|  | db5 | 96.36 | $94.39 \pm 1.45$ | 0.048 | $0.083 \pm 0.041$ |
|  | db6 | 96.15 | $88.96 \pm 10.83$ | 0.044 | $0.216 \pm 0.146$ |
|  | db7 | 97.81 | $92.65 \pm 7.80$ | 0.042 | $0.113 \pm 0.083$ |
|  | db8 | 98.21 | $89.17 \pm 8.83$ | 0.100 | $0.133 \pm 0.046$ |
|  | db9 | 97.70 | $94.51 \pm 3.46$ | 0.050 | $0.096 \pm 0.034$ |
|  | db10 | 97.81 | $90.79 \pm 7.77$ | 0.047 | $0.123 \pm 0.048$ |
|  | sym4 | 97.24 | $94.09 \pm 2.26$ | 0.057 | $0.200 \pm 0.127$ |
|  | sym5 | 97.60 | $95.57 \pm 2.58$ | 0.032 | $0.066 \pm 0.035$ |
|  | sym6 | 97.98 | $97.02 \pm 0.81$ | 0.047 | $0.172 \pm 0.095$ |
|  | sym7 | 97.16 | $92.01 \pm 7.65$ | 0.066 | $0.173 \pm 0.090$ |
|  | sym8 | 98.80 | $96.92 \pm 1.69$ | 0.062 | $0.142 \pm 0.081$ |
|  | sym9 | 98.61 | $96.34 \pm 2.12$ | 0.060 | $0.107 \pm 0.067$ |
|  | sym10 | 98.82 | $87.88 \pm 17.77$ | 0.028 | $0.142 \pm 0.115$ |
|  | coif2 | 96.23 | $87.69 \pm 9.59$ | 0.101 | $0.181 \pm 0.090$ |
|  | coif3 | 97.66 | $93.80 \pm 4.07$ | 0.043 | $0.081 \pm 0.031$ |
|  | coif4 | 98.24 | $92.30 \pm 4.80$ | 0.025 | $0.078 \pm 0.043$ |
|  | coif5 | 98.48 | $93.29 \pm 3.33$ | 0.056 | $0.217 \pm 0.178$ |
|  | coif6 | 98.82 | $89.83 \pm 15.31$ | 0.035 | $0.116 \pm 0.099$ |
|  | coif7 | 99.11 | $86.35 \pm 14.23$ | 0.028 | $0.123 \pm 0.108$ |
|  | coif8 | 98.81 | $97.69 \pm 1.26$ | 0.026 | $0.077 \pm 0.048$ |
|  | coif9 | 98.60 | $96.15 \pm 3.25$ | 0.042 | $0.101 \pm 0.047$ |
|  | coif10 | 98.47 | $96.40 \pm 2.40$ | 0.027 | $0.060 \pm 0.028$ |
| LCNN | STFT | 99.88 | $98.33 \pm 1.85$ | **0.001** | $0.019 \pm 0.018$ |
|  | sym5 | 96.89 | $95.34 \pm 1.83$ | 0.037 | $0.085 \pm 0.053$ |
| AST | STFT | 99.37 | $98.31 \pm 1.49$ | 0.007 | $\mathbf{0.018 \pm 0.016}$ |
|  | sym5 | 93.63 | $91.98 \pm 0.98$ | 0.065 | $0.081 \pm 0.010$ |
| GMM | LFCC (Frank & Schönherr) | – | – | 0.062 | – |
| RawNet2 | raw (Frank & Schönherr) | – | – | 0.363 | – |

Table 8: Detailed deepfake detection accuracies for each vocoder (generator) for the DCNN-sym5 and DCNN-coif4 for the extended wavefake experiments from Table 6.

| | Accuracy [%] | | | |
| | DCNN-sym5 | | DCNN-coif4 | |
| Generator | max | $\mu \pm \sigma$ | max | $\mu \pm \sigma$ |
|---|---|---|---|---|
| MelGAN | 98.68 | $91.59 \pm 10.22$ | 98.24 | $92.52 \pm 3.58$ |
| MelGAN (L) | 93.31 | $81.47 \pm 11.84$ | 90.23 | $77.23 \pm 7.73$ |
| HifiGAN | 100.0 | $99.07 \pm 1.32$ | 99.92 | $99.92 \pm 0.14$ |
| Multi-Band MelGAN | 98.99 | $91.63 \pm 10.17$ | 97.93 | $90.80 \pm 5.52$ |
| Full-Band MelGAN | 99.99 | $99.58 \pm 0.17$ | 100.0 | $99.21 \pm 0.73$ |
| WaveGlow | 100.0 | $100.0 \pm 0.00$ | 100.0 | $100.0 \pm 0.00$ |
| Parallel WaveGAN | 100.0 | $100.0 \pm 0.00$ | 100.0 | $100.0 \pm 0.00$ |
| Conformer | 99.99 | $99.03 \pm 0.46$ | 99.93 | $98.86 \pm 1.47$ |
| JSUT Multi-Band MelGAN | 100.0 | $99.51 \pm 0.64$ | 99.69 | $98.50 \pm 5.03$ |
| JSUT Parallel WaveGAN | 99.95 | $98.28 \pm 2.56$ | 98.96 | $80.48 \pm 18.93$ |

Table 9: Number of model parameters and filter lengths for all trained deep fake detectors. The STFT-based and WPT-based networks have the same amount of optimizable parameters and the signed-log (SG) networks use a similar amount as well. For the STFT we use a hop length of 220 samples between the windows to achieve the same input dimensions as the WPT transformed audios.

| | Model | Freq. Bins | Filter length | Model Parameters |
|---|---|---|---|---|
| DCNN | STFT | 256 | – | 239,015 |
| | haar | 256 | 2 | 239,015 |
| | db2 | 256 | 4 | 237,097 |
| | db3 | 256 | 6 | 237,097 |
| | db4 & sym4 | 256 | 8 | 237,097 |
| | db5 & sym5 | 256 | 10 | 239,015 |
| | db6 & sym6 | 256 | 12 | 239,015 |
| | coif2 | 256 | 12 | 239,015 |
| | db7 & sym7 | 256 | 14 | 239,015 |
| | db8 & sym8 | 256 | 16 | 239,015 |
| | db9 & sym9 | 256 | 18 | 241,099 |
| | coif3 | 256 | 18 | 241,099 |
| | db10 & sym10 | 256 | 20 | 241,099 |
| | coif4 | 256 | 24 | 241,099 |
| | sym14 | 256 | 28 | 243,349 |
| | coif5 | 256 | 30 | 243,349 |
| | coif6 | 256 | 36 | 245,765 |
| | coif7 | 256 | 42 | 248,347 |
| | coif8 | 256 | 48 | 248,347 |
| | coif9 | 256 | 54 | 251,095 |
| | coif10 | 256 | 60 | 254,009 |
| LCNN | sym5 | 256 | 10 | 3,312,450 |
| AST | STFT | 256 | – | 85,256,450 |
| AST | sym5 | 256 | 10 | 85,256,450 |

### 6.5   Network architecture

We list the total number of optimizable parameters in Table 9. We find a small increase in parameters in the signed scenario.

