# OpenReview forum: "Towards generalizing deep-audio fake detection networks"
_TMLR — Accepted by TMLR_

### Review · Reviewer_DGRu · 2024-01-15

**Summary Of Contributions:**

This paper primarily focuses on advancing the field of synthetic audio detection. It challenges the previously held notion that deep networks do not generalize well to unknown generators in the audio domain, a perspective notably presented in the Frank & Schönherr (2021) paper. The authors present a new approach by analyzing frequency domain fingerprints of current audio generators and training lightweight detectors that show improved generalization capabilities. They report enhanced results on the WaveFake dataset, including its extended version with additional samples from novel networks like Avocodo and BigVGAN.

**Audience:**

No

**Broader Impact Concerns:**

I don't see any discussion about broader impact. IMO, this work is for a specific audience within the audio community since this is an upgrade to Frank & Schönherr paper.

**Claims And Evidence:**

Yes

**Requested Changes:**

The entire flow of the current paper seems to be the authors reproduced the experiments done by Frank & Schönherr, and replaced the original neural network which did not perform well with the better performing DCNNs. Then the authors presented supporting results showing better performance and used ablation to demonstrate the different architecture indeed contributed.
This entire structure is easy to follow, but it would be great if the authors could spare more effort describing the motivation of using dilated convolution instead of the vanilla ones, and the results would be more convincing if the authors could list more audio examples (preferrably demos) demonstrating the cases that previous models could not classify/ whereas the proposed method could do.

**Strengths And Weaknesses:**

The novelty of the paper lies in its approach to identifying stable frequency domain artifacts across various speech synthesis networks and leveraging these findings to develop more effective detection tools.
- The success of using DCNN in this work contrast to previous studies that were more limited in their ability to generalize across different audio generators.
- The paper also incorporates newer text-to-speech synthesis networks in its analysis, contributing to a more up-to-date understanding of the domain.
However, the critical aspect of the paper is that it seems heavily focused on outperforming the Frank & Schönherr paper, potentially at the expense of broader innovative contributions to the field. While it successfully addresses the issue of generalization in deepfake audio detection, the paper could have explored more diverse methodologies or applications beyond this specific comparison.

---

> ### Author Response · Authors · 2024-01-25
> **Thank you for your feedback**
>
> Dear Reviewer DGRu,
>
> Thank you very much for your review and constructive feedback.
>
> The new revision adds extra ablation experiments in Table 4 (see below). The experimental section now also reports the performance of a dilation-free DCNN version. RawNet2 integrated information over time via recurrent connections. Section 4 additionally clarifies that we employ dilation to compensate for the removal of recurrent connections in the DCNN architecture. The ablation section argues that the additional numbers support this intuition.
>
> In addition to the generator artifacts in the `generator_artifacts` folder, the supplementary material now also contains the requested examples of misclassification. The samples in the `classification_examples` folder are misclassified without dilation and correctly identified by the dilated network. Providing all possible combinations in the supplementary material is not feasible. Consequently, we will include code, allowing users to obtain samples of interest on their own, in the final code release.
>
> Regarding a potential audience, we argue that this paper is mainly of interest to the deepfake-detection sub-community, which is well and active. The related work discussed in sections 2.2 and 2.3 provides evidence of community activity.
>
>
> **Table 4**
>
> DCNN fake-detection results on the extended WaveFake dataset (with JSUT) comparing a modiﬁed
> version of our architecture depicted in ﬁgure 3 with small modiﬁcations to motivate the use of max pooling
> and dropout. All input transforms work with 256-frequency bins. Our models are trained exclusively on
> samples drawn from a full-band MelGAN. We report the test set accuracy and aEER. To add additional
> context, we report mean test set accuracy and aEER and standard deviation over ﬁve runs for all experiments
> using seeds 0 to 4.
>
> |                    |                     | Accuracy |             |   aEER||
> |--------------------|---------------------|-------|------------------|-------|------------------|
> | DCNN-variant       | Input               | max   | µ ± σ            | min   | µ ± σ            |
> |--------------------|---------------------|-------|------------------|-------|------------------|
> ||      sym5               | 97.70 | 95.25 ± 3.09     | 0.031 | 0.069 ± 0.036    |
> ||coif4| 97.90 | 91.32 ± 5.41     | **0.025** | 0.084 ± 0.047    |
> | without Max Pooling| sym5                | 97.39 | 95.59 ± 1.73     | 0.056 | 0.082 ± 0.029    |
> | without Dropout    | sym5                | 97.75 | 95.74 ± 1.41     | 0.049 | **0.062 ± 0.014**    |
> | without Dropout    | coif4               | 97.60 | 93.25 ± 5.28     | 0.029 | 0.073 ± 0.044    |
> | without Dilation   | sym5                | 93.02 | 87.76 ± 3.21     | 0.070 | 0.117 ± 0.028    |
> | without Dilation   | coif4               | 98.46 | 81.16 ± 14.96    | 0.040 | 0.173 ± 0.094    |

---

### Review · Reviewer_KUsP · 2024-02-08

**Summary Of Contributions:**

This paper explores the use of frequency domain fingerprints, such as digital watermarks, for detecting fake audio examples. It primarily extends methodologies from related work in the image processing domain (i.e., DCNN).

The paper lacks significant algorithmic innovation, which limit its appeal to the broader TMLR community.

- Although the results offer some empirical insights, it is challenging to argue that the work contributes broad general machine learning discoveries, appearing more as an application-focused paper.

**Audience:**

No

**Broader Impact Concerns:**

I think this paper is with less interests to the TMLR community.

**Claims And Evidence:**

No

**Requested Changes:**

In addition to the DCNN, more advanced acoustic modeling architectures, such as the Audio Spectrogram Transformer, should be considered if this work primarily aims for application-oriented studies.

**Strengths And Weaknesses:**

Pros
- The authors demonstrate the effectiveness of their method on the WaveFake dataset and an extended version that includes samples from newer generators like Avocodo and BigVGAN.

Cons
- the algorithm is with less novelty and mainly incremental.

---

> ### Author Response · Authors · 2024-02-26
> **Response to reviewer KUsP**
>
> Thank you for your insightful feedback and your participation in the peer review process. We have taken your suggestion into account and updated the paper accordingly.
>
> A key goal of this paper was to reproduce the results from Frank and Schönherr to find at least one network that does not suffer from the generalization problems they observed. We believe the paper presents enough evidence for us to claim both.
>
> The review reports 'No' in the 'Claims and Evidence' field. We are struggling to understand where this answer is coming from. Our paper comes with careful evaluation. We consistently report mean and standard deviations for all numbers to ensure we do not share outliers without context. We kindly ask for an example of a statement without evidence from the paper. We are happy to address any such issue by removing the statement or providing the required evidence.
>
> In response to your suggestion regarding the consideration of different acoustic modeling architectures, such as the Audio Spectrogram Transformer (AST), we integrated the AST into our study. Specifically, we utilized the ASTModel implementation from the official GitHub repository.
>
> In our experimentation, we work with a learning rate of 0.00004, which is 10 times smaller than that typically used as suggested by the AST paper authors. We tested this approach on our dataset, including samples from newer generators like Avocodo and BigVGAN.
>
> We observe excellent performance on the original Wavefake dataset. Here, AST delivers the best average performance over multiple seeds. However, despite our efforts, we found that the transformer failed to generalize to newer generators.  In particular, we observed a lack of generalization when presented with samples from newer unknown vocoders like BigVGAN. This phenomenon persisted regardless of whether we utilized the STFT or Wavelet Packet features. In analogy with our previous findings, we found the WPT input transformation to perform better in terms of generalization to new vocoders than the STFT. Further, models like AST count more than 85M parameters while our model uses only about 200K.
>
> We have summarized our findings in the tables below:
> Fake detection results on the extended WaveFake dataset including JSUT.  All models are trained exclusively on samples drawn from a full-band MelGAN. We report the test set accuracy and aEER.
>
> **Extended WaveFake**
> | Network | Input | Accuracy [%] | Accuracy Mean ± Std | aEER | aEER Mean ± Std |
> |---------|-------|--------------|---------------------|------|------------------|
> | DCNN (ours)   | STFT  | 96.46        | 91.72 ± 2.94        | 0.036| 0.159 ± 0.150    |
> |         | sym5  | 97.70        | 95.25 ± 3.09        | 0.031| 0.069 ± 0.036    |
> | AST     | STFT  | 90.98        | 87.10 ± 2.54        | 0.089| 0.122 ± 0.021    |
> |         | sym5  | 93.49 | 91.25 ± 1.38 | 0.065 | 0.087 ± 0.013 |
>
> **WaveFake**
> | Network | Input | Accuracy [%] | Accuracy Mean ± Std | aEER | aEER Mean ± Std |
> |---------|-------|--------------|---------------------|------|------------------|
> | DCNN (ours)   | STFT  | 99.88        | 97.98 ± 3.18        | 0.001| 0.099 ± 0.178    |
> |         | sym5  | 97.60        | 95.57 ± 2.58        | 0.032| 0.066 ± 0.035    |
> | AST     | STFT  | 99.37        | 98.31 ± 1.49        | 0.007| 0.018 ± 0.016    |
> |         | sym5  | 93.63   | 91.98 ± 0.98    | 0.065 | 0.081 ± 0.010 |
>
> Results on BigVGAN and Avocodo. When looking at the two newer generators in isolation, we observe aEERs in line with previous results for our wavelets-based classifiers. The AST and the networks with STFT inputs drop in performance when evaluated with the unseen BigVGAN.
>
> **Avocodo**
> | Network | Input | Accuracy [%] | Accuracy Mean ± Std | aEER | aEER Mean ± Std  |
> |---------|-------|--------------|---------------------|------|------------------|
> | DCNN (ours)   | STFT  | 99.99        | 91.40 ± 16.49       | 0.001| 0.096 ± 0.179    |
> |         | sym5  | 99.34        | 97.48 ± 2.21        | 0.010| 0.046 ± 0.039    |
> | AST     | STFT  | 99.61        | 98.84 ± 0.62        | 0.005| 0.021 ± 0.012    |
> |         | sym5  | 98.18   | 97.86 ± 0.28    | 0.023 | 0.028 ± 0.004    |
>
> **BigVGAN**
> | Network | Input | Accuracy [%] | Accuracy Mean ± Std | aEER | aEER Mean ± Std |
> |---------|-------|--------------|---------------------|------|-----------------|
> | DCNN (ours)   | STFT  | 84.67        | 65.12 ± 11.82       | 0.182 | 0.365 ± 0.095  |
> |         | sym5  | 96.28        | 92.66 ± 4.02        | 0.043 | 0.097 ± 0.051  |
> | AST     | STFT  | 62.95        | 50.41 ± 7.48        | 0.357 | 0.424 ± 0.039  |
> |         | sym5  | 91.99   |   88.53 ± 2.36  |   0.095   | 0.139 ± 0.029  |
>
> The first results are promising. Further research is necessary to enhance AST's ability to generalize to unseen sources (e.g. BigVGAN). However, since this paper's original version made no claims involving AST, we feel that such future endeavors are beyond the scope of this rebuttal.

---

### Review · Reviewer_3aDJ · 2024-02-12

**Summary Of Contributions:**

This paper proposes to extend the Wavefake dataset with two additional Text-to-Speech systems, and also proposes a 2D convolutional network on the generated voice detection problem. They show with experimental results that this proposed network works slightly better than the other existing networks on the same dataset.

**Audience:**

Yes

**Broader Impact Concerns:**

The subject area of this paper is in fact beneficial socially in terms of societal effects, as it intends to accurately detect from AI generated content.

**Claims And Evidence:**

Yes

**Requested Changes:**

- I think it is very important to justify what are the reasons for choosing the baselines methods LCNN as GMM.
- It would help the paper significantly if the authors could analyze the performance with respect to different TTS systems. You are doing that for BigVCan and Avocodo, but it would help to include comparisons with respect to other TTS systems.
- Why are you not comparing with LCNN in table 3?

**Strengths And Weaknesses:**

Strengths:
- The proposed network seems to work better than the other methods the authors compared against. However I am not sure if the list of methods that they have compared against includes most up-to-date methods. In fact it only includes two other methods, and one of which is a simple GMM. So I am not entirely sure of the validity of the claim of the paper the proposed method outperforms other alternatives.
- The paper provides several ablation studies on several hyperparameters.
- The comparison of average frequency profiles between original LJSpeech recordings and the generated ones is interesting.

Weaknesses:
- I am not sure if the paper really proposes something to advance 'generalization' of detecting generated voice. It proposes to use a convolutional network, which it shows that it works reasonably well. However, there is no novel methodology to advance our understanding towards building a singular network that better generalizes compared to straightforward techniques.
- The paper shows an extensive effort for analyzing the effect of using different types of wavelets. However these wavelet transforms are only applied for the proposed method, and not for the baseline method LCNN (e.g. Table 1). This somehow add uncertainty as to whether the performance improvement is due to the proposed CNN method or the input transform.
- Overall, I am not sure what is the contribution of the paper, because of the two weaknesses I have listed above.

---

> ### Author Response · Authors · 2024-02-23
> **Thank you for reviewing our work, we have made the requested changes.**
>
> Dear 3aDJ,
>
> we thank you for taking the time to read and review our work.
>
> Our choices of the Gaussian mixture model (GMM) and the light convolutional neural network (LCNN) baselines follow the related work. Frank & Schönherr reported that the Gaussian mixture model performed best on the original wavefake data set. The LCNN is the leading network in the ASVspoof 2021 challenge.
> Consequently, our baseline selection is well-founded based on the related work.
>
> We added results for the LCNN trained on STFT and sym5 features to tables 2 and 3 (see below). The LCNN struggles to identify BigVGAN samples correctly, which causes the overall performance differences.
> Furthermore, Table 8 (see below) now lists recognition accuracies for all remaining generators, excluding Avocodo and BigVGAN, which appear in Table 3. Our DCNN remains the best-performing network on the extended data set, however we do not repeat Table 1 here.
>
> **Table 2**
>
> Results on the unmodiﬁed WaveFake dataset (Frank & Schönherr, 2021). We cite baseline numbers
> as reported in the original paper. All frequency representations work with 256 bins. Our network outperforms the GMM-LFCC and their RawNet2 implementation for the db4, db5, sym4, sym5, coif4 as well as the coif8 wavelet, and the STFT.
>
> |Network|Input|Accuracy[%] max|Accuracy µ ± σ|aEER min|aEER µ ± σ|
> |------|-----|---------|-------------|----|-------------|
> |DCNN(ours)|STFT|99.88|97.98 ± 3.18|**0.001**|0.099 ± 0.178|
> ||db4|94.71|90.20 ± 5.90|0.079|0.136 ± 0.050|
> ||db5|96.36|94.39 ± 1.45|0.048|0.083 ± 0.041|
> ||db8|98.21|89.17 ± 8.83|0.100|0.133 ± 0.046|
> ||sym4|97.24|94.09 ± 2.26|0.057|0.200 ± 0.127|
> ||sym5|97.60|95.57 ± 2.58|0.032|0.066 ± 0.035|
> ||sym8|98.80|96.92 ± 1.69|0.062|0.142 ± 0.081|
> ||coif4|98.24|92.88 ± 5.05|0.025|0.118 ± 0.088|
> ||coif8|98.81|97.87 ± 0.92|0.026|0.121 ± 0.089|
> |LCNN|STFT|99.88|98.33 ± 1.85|**0.001**|0.019 ± 0.018|
> ||sym5|96.89|95.34 ± 1.83|0.037|0.085 ± 0.053|
> |AST|STFT|99.37|98.31 ± 1.49|0.007|**0.018 ± 0.016**|
> ||sym5|93.63|91.98 ± 0.98|0.065|0.081 ± 0.010|
> |GMM|LFCC|-|-|0.062|-|
> |RawNet2|raw|-|-|0.363|-|
>
> **Table 3**
>
> Results on BigVGAN (Lee et al., 2023a) and Avocodo (Bak et al., 2022). When looking at the two
> newer generators in isolation, we observe aEERs in line with previous results for our wavelets-based classiﬁers.
> The GMM-LFCC (Frank & Schönherr, 2021) and the networks with STFT-inputs drop in performance when
> evaluated with the unseen BigVGAN.
>
> | ||Avocado | | | |BigVGAN(L)| | | |
> |------|------|---------|------|------|------|-------------|------|------|------|
> | | |Accuracy[%]||aEER||Accuracy[%]||aEER||
> |Network|Input|max|µ ± σ|min|µ ± σ|max|µ ± σ|min|µ ± σ|
> |------|------|---------|---------|------|---------|---------|---------|------|---------|
> |DCNN(ours)|STFT|99.99|91.40 ± 16.49|**0.001**|0.096 ± 0.179|84.67|65.12 ± 11.82|0.182|0.365 ± 0.095|
> ||db4|98.46|94.11 ± 3.69|0.024|0.098 ± 0.056|93.78|89.68 ± 2.92|0.079|0.136 ± 0.047|
> ||db5|98.60|96.13 ± 2.87|0.026|0.064 ± 0.048|97.47|93.31 ± 3.09|0.052|**0.097 ± 0.047**|
> ||db8|99.53|96.92 ± 2.23|0.005|0.055 ± 0.041|94.88|87.77 ± 6.81|0.101|0.159 ± 0.061|
> ||sym4|97.85|86.58 ± 11.73|0.033|0.184 ± 0.142|93.58|86.55 ± 5.57|0.094|0.226 ± 0.108|
> ||sym5|99.34|97.48 ± 2.21|0.010|0.046 ± 0.039|96.28|92.66 ± 4.02|**0.043**|**0.097 ± 0.051**|
> ||sym8|99.76|91.82 ± 6.43|0.003|0.130 ± 0.095|95.08|90.85 ± 3.69|0.064|0.166 ± 0.071|
> ||coif4|99.72|95.24 ± 7.07|0.005|0.072 ± 0.101|95.94|87.23 ± 7.40|0.047|0.173 ± 0.093|
> ||coif8|98.70|92.31 ± 6.58|0.025|0.122 ± 0.090|96.11|93.46 ± 3.32|0.058|0.128 ± 0.083|
> |LCNN|STFT|99.89|99.67 ± 0.19|**0.001**|**0.006 ± 0.004**|70.32|57.71 ± 10.42|0.308|0.382 ± 0.060|
> ||sym5|98.91|95.81 ± 3.66|0.011|0.069 ± 0.060|93.76|92.05 ± 1.45|0.069|0.112 ± 0.042|
> |AST|STFT|99.61|98.84 ± 0.62|0.005|0.021 ± 0.012|62.95|50.41 ± 7.48|0.357|0.424 ± 0.039|
> ||sym5|98.18|97.86 ± 0.28|0.023|0.028 ± 0.004|91.99|88.53 ± 2.36|0.095|0.139 ± 0.029|
> |GMM|LFCC|–|–|0.316|–|–|–|0.432|–|
>
>
> **Table 8**
>
> Detailed deepfake detection accuracies for each vocoder (generator) for the DCNN-sym5 and DCNN-
> coif4 for the extended wavefake experiments from Table 6.
>
> | |DCNN-sym5|DCNN-sym5|DCNN-coif4|DCNN-coif4|
> |-------------------|----------|----------------|----------|----------------|
> | Generator|Acc max|Acc µ ± σ|Acc max|Acc µ ± σ|
> |-------------------|----------|----------------|----------|----------------|
> |MelGAN|98.68|91.59 ± 10.22|98.24|92.52 ± 3.58|
> |MelGAN (L)|93.31|81.47 ± 11.84|90.23|77.23 ± 7.73|
> |HiFiGAN|100.0|99.07 ± 1.32|99.92|99.92 ± 0.14|
> |Multi-Band MelGAN|98.99|91.63 ± 10.17|97.93|90.80 ± 5.52|
> |Full-Band MelGAN|99.99|99.58 ± 0.17|100.0|99.21 ± 0.73|
> |WaveGlow|100.0|100.0 ± 0.00|100.0|100.0 ± 0.00|
> |Parallel WaveGAN|100.0|100.0 ± 0.00|100.0|100.0 ± 0.00|
> |Conformer|99.99|99.03 ± 0.46|99.93|98.86 ± 1.47|
> |JSUT Multi-Band MelGAN|100.0|99.51 ± 0.64|99.69|98.50 ± 5.03|
> |JSUT Parallel WaveGAN|99.95|98.28 ± 2.56|98.96|80.48 ± 18.93|

---

### Author Response · Authors · 2024-03-08
**End of discussion period**

Dear Reviewers,

Our discussion will come to an end in three days. We thank everyone for reading our paper and for their instructive feedback. Naturally, important issues have been raised. In response, we have added, among other changes, a dilation ablation experiment (DGRu), evaluated the Audio Spectrogram Transformer (KUsP) and added additional details related to the performance of the light CNN (3aDJ).

We believe this paper is of interest to the active deepfake-detection subcommunity. We also hope readers will appreciate our open-source code and the dataset extension. To maintain our anonymity, the action editor has confidential access to both.

Should you have further questions, feel free to reach out.
Thank you very much for your efforts,

The paper 1969 authors

---

### Decision · Action_Editor_ofzr · 2024-03-26

**Recommendation:** Accept with minor revision

**Comment:**

Three reviewers give reasonable review comments and the authors also respond actively. All the reviewers acknowledge the efforts that the authors made to solve the issues raised. Although some claims lack enough evidence in the submission version, the authors have provided enough intuitive and empirical results to support the claims. All the reviewers have some concerns about the novelty of this paper. However, according to TMLR review policy, novelty should not be a criterion to reject a paper. Given these considerations, especially the authors provide good empirical evidence in the rebuttal, I recommend an acceptance to this paper. The authors should improve this paper by taking the reviewer's feedback into consideration.

**Audience:**

The paper lacks significant algorithmic innovation, which limits its appeal to the broader TMLR community. However, a specific audience within the audio community, especially the deepfake-detection subcommunity could have interests.

**Claims And Evidence:**

Reviewers point out several questions about this paper, such as 1) why choose LCNN as the baseline; 2) lack the performance with respect to different TTS systems; 3)  more advanced acoustic modeling architectures, such as the Audio Spectrogram Transformer, should be considered; 4) lack the motivation of using dilated convolution. The authors actively address these questions with either intuitive analyses or extensive empirical results. Therefore, the claims in the paper can be well supported by evidence after the rebuttal.

---

> ### Author Response · Authors · 2024-04-08
> **Camera ready version available**
>
> We would like to thank everyone for their feedback. We uploaded a new version of the paper which includes the results of our discussion. Furthermore, we also added a link to the open-source code.